# Landscape, Soil, Lithology, Climate and Permafrost Control on Dissolved Carbon, Major and Trace Elements in the Ob River, Western Siberia

**Iurii Kolesnichenko [1], Larisa G. Kolesnichenko [1], Sergey N. Vorobyev [1], Liudmila S. Shirokova [2,3], Igor P. Semiletov [4,5,6,7], Oleg V. Dudarev [4], Rostislav S. Vorobev [1], Uliana Shavrina [1], Sergey N. Kirpotin [1] and Oleg S. Pokrovsky [1,2,3,*]**

1   BIO-GEO-CLIM Laboratory, Tomsk State University, Lenin pr., 36, 634050 Tomsk, Russia; vancansywork@mail.ru (I.K.); klg77777@mail.ru (L.G.K.); soil@green.tsu.ru (S.N.V.); tsu@erd.su (R.S.V.); ulyanashavrina@yandex.ru (U.S.); kirp@mail.tsu.ru (S.N.K.)
2   Geosciences and Environment Toulouse, UMR 5563 CNRS, 14 Avenue Edouard Belin, 31400 Toulouse, France; Liudmila.SHIROKOVA@Get.omp.eu
3   N. Laverov Federal Center for Integrated Arctic Research, Russian Academy of Sciences, 163069 Arkhangelsk, Russia
4   Laboratory of Arctic Research, V.I. Il'ichev Pacific Oceanological Institute, Far Eastern Branch of Russian Academy of Sciences, 43 Baltic Street, 690041 Vladivostok, Russia; ipsemiletov@alaska.edu (I.P.S.); dudarev@poi.dvo.ru (O.V.D.)
5   Tomsk Polytechnic University, 634050 Tomsk, Russia
6   Tomsk State University, 634050 Tomsk, Russia
7   Department of Chemistry, Moscow State University, 119192 Moscow, Russia
*   Correspondence: oleg.pokrovsky@get.omp.eu

**Abstract:** In order to foresee possible changes in the elementary composition of Arctic river waters, complex studies with extensive spatial coverage, including gradients in climate and landscape parameters, are needed. Here, we used the unique position of the Ob River, draining through the vast partially frozen peatlands of the western Siberia Lowland and encompassing a sizable gradient of climate, permafrost, vegetation, soils and Quaternary deposits, to assess a snap-shot (8–23 July 2016) concentration of all major and trace elements in the main stem (~3000 km transect from the Tom River confluence in the south to Salekhard in the north) and its 11 tributaries. During the studied period, corresponding to the end of the spring flood-summer baseflow, there was a systematic decrease, from the south to the north, of Dissolved Inorganic Carbon (DIC), Specific Conductivity, Ca and some labile trace elements (Mo, W and U). In contrast, Dissolved Organic Carbon (DOC), Fe, P, divalent metals (Mn, Ni, Cu, Co and Pb) and low mobile trace elements (Y, Nb, REEs, Ti, Zr, Hf and Th) sizably increased their concentration northward. The observed latitudinal pattern in element concentrations can be explained by progressive disconnection of groundwaters from the main river and its tributaries due to a northward increase in the permafrost coverage. A northward increase in bog versus forest coverage and an increase in DOC and Fe export enhanced the mobilization of insoluble, low mobile elements which were present in organo-ferric colloids (1 kDa—0.45 μm), as confirmed by an in-situ dialysis size fractionation procedure. The chemical composition of the sampled mainstream and tributaries demonstrated significant ($p < 0.01$) control of latitude of the sampling point; permafrost coverage; proportion of bogs, lakes and floodplain coverage and lacustrine and fluvio-glacial Quaternary deposits of the watershed. This impact was mostly pronounced on DOC, Fe, P, divalent metals (Mn, Co, Ni, Cu and Pb), Rb and low mobile lithogenic trace elements (Al, Ti, Cr, Y, Zr, Nb, REEs, Hf and Th). The pH and concentrations of soluble, highly mobile elements (DIC, $SO_4$, Ca, Sr, Ba, Mo, Sb, W and U) positively correlated with the proportion of forest, loesses, eluvial, eolian, and fluvial Quaternary deposits on the watershed. Consistent with these correlations, a Principal Component Analysis demonstrated two main factors explaining the variability of major and trace element concentration in the Ob River main stem and tributaries. The DOC, Fe, divalent metals and trivalent and tetravalent trace elements were presumably controlled by a northward increase in permafrost, floodplain, bogs, lakes and lacustrine

deposits on the watersheds. The DIC and labile alkaline-earth metals, oxyanions (Mo, Sb and W) and U were impacted by southward-dominating forest coverage, loesses and eluvial and fertile soils. Assuming that climate warming in the WSL will lead to a northward shift of the forest and permafrost boundaries, a "substituting space for time" approach predicts a future increase in the concentration of DIC and labile major and trace elements and a decrease of the transport of DOC and low soluble trace metals in the form of colloids in the main stem of the Ob River. Overall, seasonally-resolved transect studies of large riverine systems of western Siberia are needed to assess the hydrochemical response of this environmentally-important territory to on-going climate change.

**Keywords:** river; forest; bog; permafrost; carbon; major ions; iron; colloids; trace element

## 1. Introduction

Studies on hydrochemistry of large Arctic rivers are at the forefront of climate warming research due to their high importance in carbon (C) and greenhouse gases (GHG) regulation at the planetary scale and their high vulnerability to ongoing environmental changes [1,2]. Presently, researchers have achieved a satisfactory understanding of hydrological fluxes and river water hydrochemistry, including both suspended and particulate load, at the terminal (gauged) stations across the Arctic. This was possible thanks to the systematic work of the State Hydrological Surveys of the main Artic countries [3] and, more recently, concerted works of various international programs [4–9]. In contrast, the knowledge of spatial variations of major and trace components of the river water along the main stem of Arctic rivers and their tributaries, which is necessary for understanding the environmental controls and export mechanisms of riverine solutes, remains rather limited.

The Ob River, which is the largest Arctic river in terms of its watershed area (2,975,000 km$^2$), is an important vector of carbon, nutrients and major and trace element transfer to the Kara Sea [10,11]. It drains highly vulnerable discontinuous and sporadic permafrost (20% in average), which is extremely rich in organic C (OC) due to the dominance of peat soils [12]. Most recent hydrological studies demonstrated that, over past 80 years, the Ob River discharge increased by ca 7.7% [13] at a rate of 384 and 173 m$^3$/s (10 year$^{-1}$) in spring and winter, respectively, which was linked to a rapid increase in both warming and wetting of the territory [14]. This, together with its unique geographical situation and landscape setting, render the Ob River watershed among the key targeting regions for biogeochemical studies in the Arctic [2]. Indeed, the number of publications per year on the Ob River has increased from <10 prior to 1995 to 10–20 in 1996–2014 and 50–80 over the past 6 years.

Several detailed studies of Dissolved Organic Carbon (DOC) and major elements were conducted at the terminal gauging station of the river, near the Salekhard city [4–6]. These included DOC time-series observations by molecular-level techniques [15] and via remote sensing [16] and the quantification of particulate organic matter export [7]. In contrast, spatial coverage of the river main stem and its tributaries remains rather low, with just a few studies of the dissolved carbon and related CO$_2$ and CH$_4$ emissions [17,18] and one study of the molecular composition of DOC [19]. A large amount of data is available from the systematic State Rosgidromet monitoring of OC and major ions on four gauging stations of the Ob River (Salekhard, Belogor'e, Aleksandrovskoe and Kolpashevo) during 1970–2010 and measurements by Tomsk Politechnical University, as summarized in ref. [20]. The OC-controlled export of Hg by the Ob River has been considered by Mu et al. [21] and Sonke et al. [22]. A snapshot study of $^{137}$Cs was performed at the scale of the Ob Basin [23]. Much less is known about other major and trace elements, especially their variations among the tributaries.

An important feature of the Ob River basin is the dominance of wetlands and mires, which contain huge amounts of OC and provide a sizable input of DOM and relevant metals, such as Fe, to the Ob River main stem and tributaries due to strong hydrological connectivity (i.e., [24–26]). As a result of this enhanced input of Fe and DOC to the Ob

River, its waters are likely to contain a high concentration of colloids. However, this aspect has never been tested for the Ob River, although organic and organo-mineral colloids can be important carriers of a number of trace elements, as is known in small rivers [27] and surface waters [28,29] across western Siberia.

The novelty of the present work is to acquire a snap-shot picture of the DOC and major and trace elements (TE), including their colloidal forms, in the main stem and several tributaries of the Ob River and to test, for the first time for this huge territory, a landscape control, including vegetation, type of soil and Quaternary deposits, on the chemical composition of the river water. Testing the landscape control on riverine solutes is now possible due to significant progress in digitalizing the available vegetation, permafrost and climate maps of large territories that allows straightforward landscape-based interpretation of river water chemistry (see examples in [30–32]). In this study, we hypothesized a dominant control of bog and permafrost coverage on the concentrations of DOC and major and trace elements in the main stem. In particular, in accord with previous studies of small WSL rivers [10,30,31,33–36], we expected a northward increase in DOC and a decrease in soluble alkaline-earth metal concentrations. In order to reveal the mechanisms of element transport in the Ob River and its tributaries, we assessed the colloidal and truly dissolved (low molecular weight) concentration of carbon and all major and trace elements in selected samples. We hypothesized a change in organic colloids after the confluence of the Ob and Irtysh rivers, following a recent study of DOM pattern in the Ob River main stem [19]. We anticipate that acquiring this new knowledge on riverine major and trace elements over the large climate, permafrost and vegetation gradient of the Ob River basin will allow empirical but straightforward prediction of future changes in river water chemistry, following a well-established "substituting space for time" approach.

## 2. Study Site and Methods

### 2.1. The Ob River of the WSL and Its Sampling

The Ob River, which delivers 15% of the total freshwater flow to the Arctic Ocean (404 $km^3 \cdot year^{-1}$), combines the features of southern rivers, draining steppe and forest-steppe regions, via its longest tributary, the Irtysh River, and the western Siberia Lowland rivers draining through a mixture of forest and mires. Further in the north, the Ob River is influenced by permafrost peatlands. The largest tributary of the Ob River is the Irtysh, which drains a territory of 1,643,000 $km^2$ and exhibits an annual discharge of 88.371 $km^3$, which is 37.5% of that of the Ob River upstream at their confluence at Khanty-Mansiisk. During the open water high flow period of 2016, the contribution of the Irtysh amounted to 49% of the Ob River discharge upstream of the confluence.

In this work, we used ship ('OM-341' vessel, see ref. [19]) route sampling and collected 23 main stem and 6 secondary channel water samples going from the north to the south between 8 July 2016 and 23 July 2016 (Figure 1). We covered, in total, 2952 km of the river length, which encompassed a sizable gradient in the latitude (from 66°47′19″ N in the most northern site, Salemal, to 56°54′24″ N in the most southern site, Kozjulyno, Supplementary Material Table S1). The 23 sampling points of the main stem are sufficiently representative to cover the full variability of hydrochemical parameters of the river during the studied season. This is supported by the highly homogeneous physio-geographical setting of the WSL, with minimal variations in runoff, lithology, vegetation and anthropogenic pressure over the quite large north–south gradient (i.e., [29–36]), as also reflected in the quite smooth pattern of the $CO_2$ concentration in the Ob River across the same latitudinal gradient [18]. In addition to the main stem, 11 tributaries of the Ob River were collected. The summer of 2016 was rather cold (−3.1° below normal (defined as prior to 2000) for July) and slightly humid (130% of the normal precipitation for July).

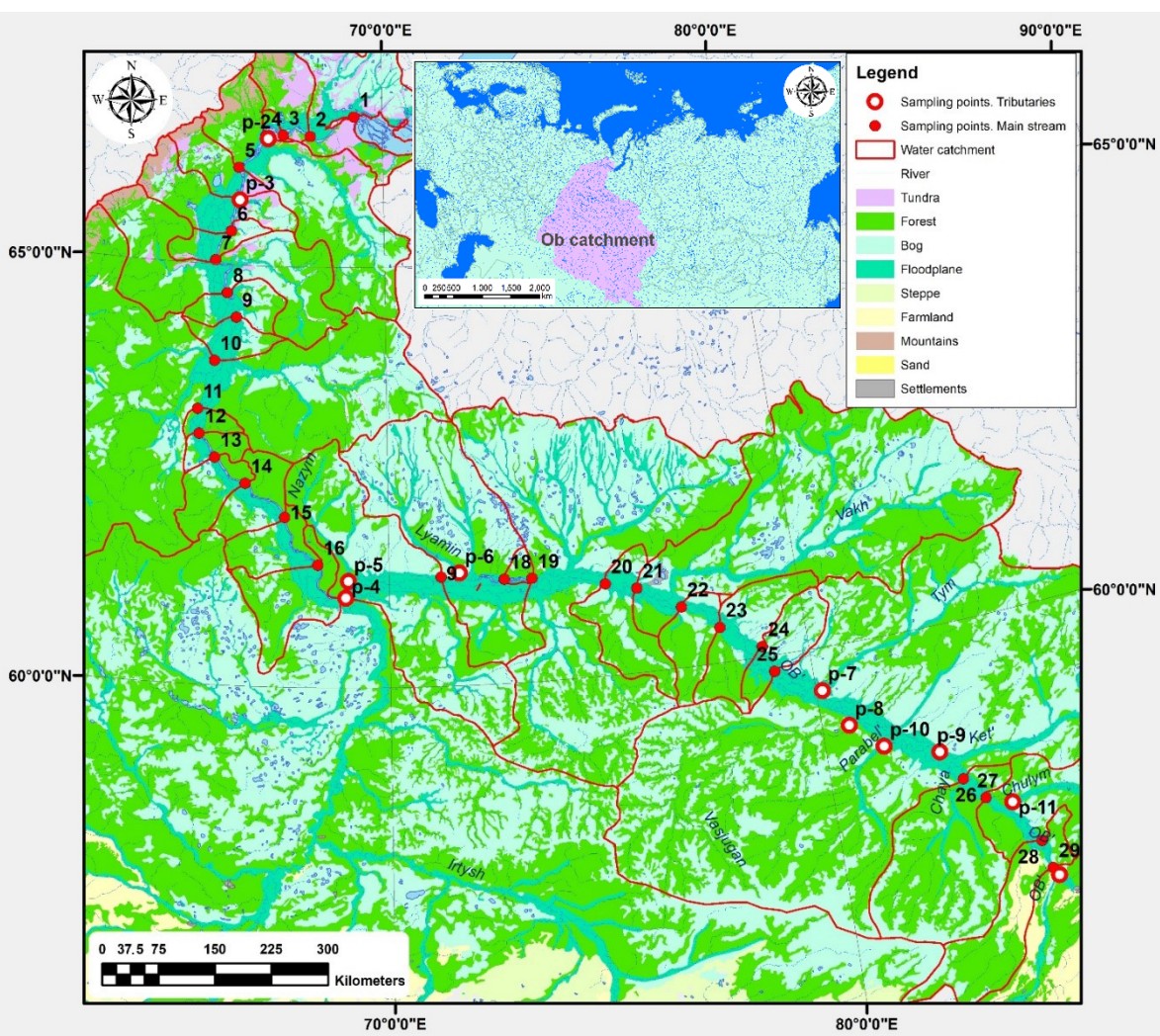

**Figure 1.** The Ob River and tributaries sampling points (solid and open circles, respectively) showing the dominant landscapes of the Ob River basin.

The river water was collected from the surface (0.5 m depth) in the central part of the main stem or a minimum of 500 m upstream of the tributary via a pre-cleaned polypropylene 1 L container and immediately filtered (<0.45 μm regenerated cellulose filter) using a pre-cleaned 250 mL polysulfone Nalgene filter unit, combined with a vacuum pump. First, 250 mL of MilliQ water was filtered, and the first portion of the river water filtrate (250 mL) was discarded. There was no decrease in the rate of filtration for the sampled volumes (typically less than 500 mL), so we did not expect any artifacts linked to filter clogging.

In addition to traditional 0.45 μm filtration, 10 selected samples of the main stem and tributaries were processed for 1 kDa (~0.0013 nm) dialysis. This technique allows one to quantify the nominal low molecular weight ($LMW_{<1kDa}$) and colloidal (1 kDa—0.45 μm) fractions. For this, large volumes of the river water were collected into a thoroughly cleaned 5 L plastic container via filtration through a 20 μm Nylon net, to avoid large particles, zooplankton and insects. Dialysis experiments were performed using 50 mL pre-cleaned dialysis bags placed in the river water over 3 to 5 days, as described elsewhere [28,37]. The plastic container was kept in darkness at a temperature similar to that of the river water and gently agitated due to ship movement and manually. The stability of the river water chemical composition during the full length of the dialysis procedure was verified by comparison of the dissolved (<0.45 μm) concentrations of all solutes before and after the exposure; the difference did not exceed 10% (at $p < 0.01$). As such, even if some microbial

activity could occur during dialysis, it did not modify the colloidal composition of the river water.

### 2.2. Analytical Techniques

All filtered and dialyzed samples were stored in the refrigerator for 1 month prior to the analyses. Major anion concentrations ($Cl^-$ and $SO_4{}^{2-}$) were measured by ion chromatography (Dionex 2000i) with an uncertainty of 2%. The dissolved organic carbon (DOC) and dissolved inorganic carbon (DIC) was determined using a Shimadzu TOC-VSCN Analyzer with an uncertainty of 3% and a detection limit of 0.3 mg/L [38]. In samples with low DOC (<10 mg/L), the DIC was also analyzed via a potentiometric alkalinity titration procedure with an uncertainty of $\pm 1\%$ and a detection limit of $5 \times 10^{-5}$ mol $L^{-1}$; the difference between the results of the Shimadzu did not exceed 5%.

Major and trace elements were measured by quadrupole ICP-MS (7500ce, Agilent Technologies). Indium and rhenium were used as internal standards at their concentrations of approximately 3 μg $L^{-1}$. The international geostandard SLRS-5 (Riverine Water Reference Material for Trace Metals certified by the National Research Council of Canada) was used to check the validity and reproducibility of each analysis (see ref. [37] for analytical details). There was good agreement between our replicated measurements of SLRS-5 and the certified values (relative difference < 15%). Elements lacking the certified values in the SLRS sample or those having high intrinsic uncertainty of measurements (>20%) are not discussed in this work (Be, Sn and Te). For all major and most trace elements, analyzed by ICP MS, the concentrations in the blanks were below the analytical detection limits ($\leq$0.1–1 ng/L for Cd, Ba, Y, Zr, Nb, REE, Hf, Pb, Th and U; 1 ng/L for Ga, Ge, Rb, Sr and Sb; ~10 ng/L for Ti, V, Cr, Mn, Fe, Co, Ni, Cu, Zn and As). Further details of analyses are provided elsewhere [27,28,31].

### 2.3. Landscape Parameters of Tributaries and the Main Stem

The landscape parameters of the 9 tributaries and 23 points of the Ob River main stem were determined by digitalizing available soil, vegetation, lithological and geocryological maps (Figure 1, Table S2). The landscape and soil parameters were typified using the United States Database of Soil Resources (Available online: http://egrpr.soil.msu.ru/, accessed on 7 November 2021); the borders were verified and corrected according to available Landsat images. The permafrost parameters of the watershed were obtained from NCSCD data. The type of Quaternary deposits was taken from the State Geological Map of Russia with a resolution of 1:1,000,000 (Available online: http://www.geolkarta.ru/, accessed on 7 November 2021).

The Spearman rank order correlation coefficient (Rs) ($p < 0.05$) was used to determine the relationship between each major and trace element concentration and the latitude, climatic, lithological and landscape parameters of the watersheds. Further statistical treatment of element concentration drivers in river waters included a Principal Component Analysis with a variance estimation method and a scree test to minimize the number of governing factors. This analysis allowed us to test the effect of various environmental parameters (landscape, soil, vegetation, permafrost and type of Quaternary deposits) of the watershed on spatial variations of riverine solutes in both the Ob River main stem and the tributaries.

## 3. Results

### 3.1. Impact of the Latitude on the Element Concentration in the Main Stem of the Ob River

According to the major and trace element behavior along the water course of the main stem [39], from the south to the north transect, three main families were distinguished, as illustrated in Figures 2 and 3. The Dissolved Inorganic Carbon, alkaline-earth metals (Ca and Sr), Mo, W and U sizably (by a factor of 2.0) decreased their concentrations northward (Figure 2).

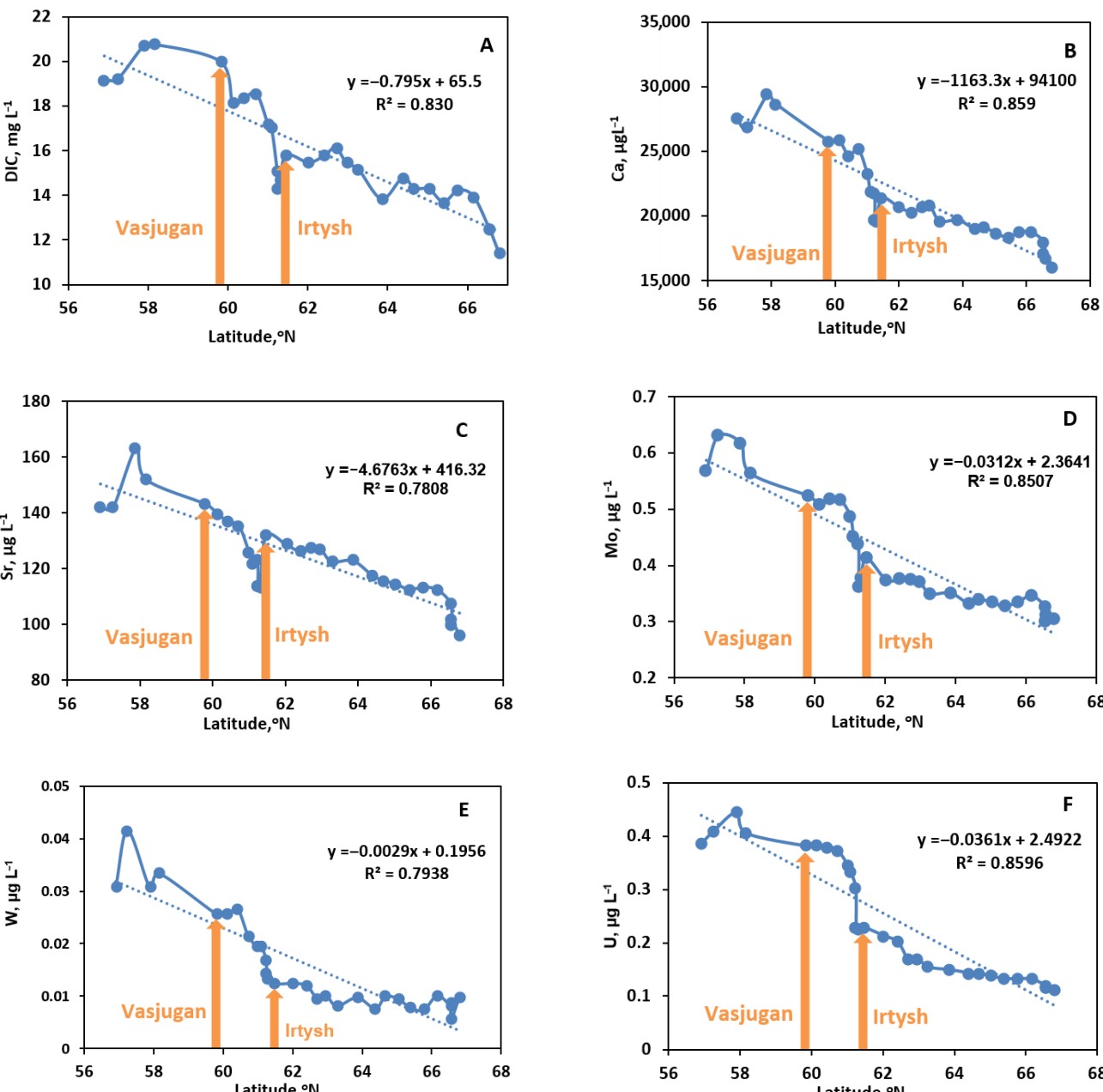

**Figure 2.** Latitudinal trends of labile major and trace elements decreasing their concentrations in the main stem of the Ob River from the south to the north: DIC (**A**), Ca (**B**), Sr (**C**), Mo (**D**), W (**E**) and U (**F**). The two main tributaries likely to control DOC and trace metal delivery, rivers Vasjugan and Irtysh, are shown by arrows.

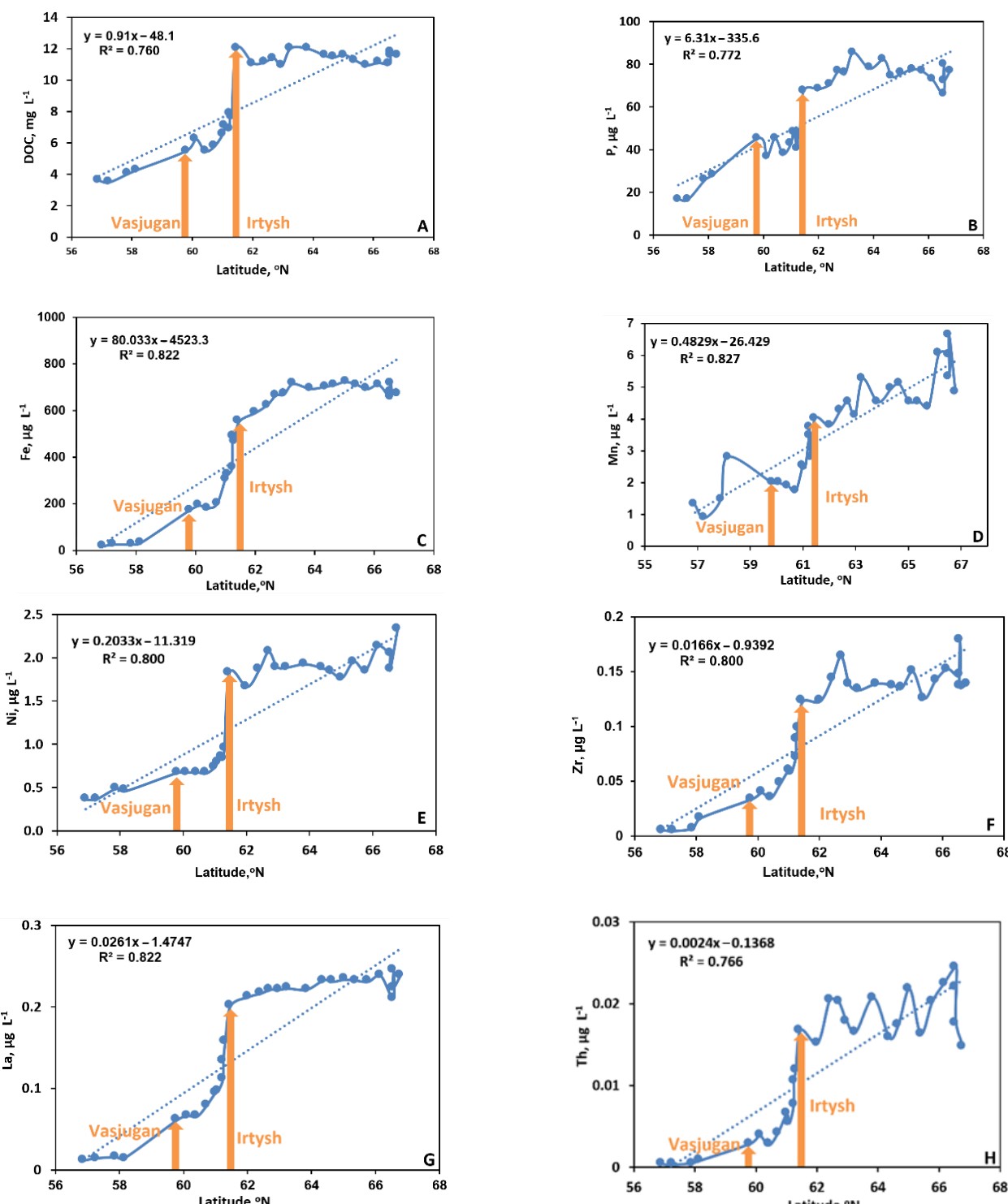

**Figure 3.** Latitudinal trends of DOC (**A**), nutrients (P (**B**), Fe (**C**), Mn (**D**) and Ni (**E**)) and insoluble trace elements (Zr (**F**), La (**G**) and Th (**H**)), showing a strong increase in concentration in the Ob main stem after the confluence with the Vasyugan and Irtysh River (shown by arrows).

The second group of elements included DOC, P, Mn, Al, Fe, Ti, Ni, Cu, Co, Rb, Pb, Y, Zr, Nb, REEs, Hf and Th, which strongly (by a factor of 2 to 10) increased their concentration from the south to the north (Figure 3). An increase in these element concentrations in the main stem of the Ob River occurred after the confluence, first, with Vasjugan and, then, with Irtysh; the impact of the Irtysh was especially seen for major anions (Cl and $SO_4$),

alkali (Na and K) and Pb (Figure S1 of the Supplementary Material). Finally, a group of elements did not exhibit any sizable (>1.5 x) change in concentration over the main stem (Si, Li, V and Ba), or the evolution of their concentration did not follow any specific pattern (B, Mg, Cr, Zn, Cd, Ga, Ge, Cs and Tl), as illustrated in Figure S2. Note, some peaks in concentrations were detected for Zn, Cd and Pb, which could be tentatively linked to local pollution sources.

One potentially important tributary of the Ob River is the Vasyugan River. This tributary drains the largest world mire [40], and thus, it can substantially modify the Ob River chemical composition via delivering a number of DOM-bound elements. To assess this impact, the degree of element enrichment in the northern part of the Ob River (downstream of the Irtysh) was compared to its intermediate part (between Vasyugan and Irtysh) and the southern part (between Tom and Vasyugan). For this, we calculated the average concentrations of DOC and major and trace elements in these three segments of the main stem (Table 1). It can be seen that the elements mostly enriched (by a factor of 2 to 10) in the northern part of the river were DOC, Cl, P, Mn, Fe, some divalent metals (Mn, Ni and Pb) and insoluble lithogenic elements (Al, Ti, Nb, Y, REE, Ti, Zr, Hf and Th).

**Table 1.** Mean (±SD) concentration ($\mu g\ L^{-1}$) of major and trace elements in the three distinct parts of the Ob River main stem upstream and downstream of its confluence with Vasyugan and Irtysh.

| Descriptions | Tom–Vasyugan (*n* = 4) | Vasyugan–Irtysh (*n* = 11) | Irtysh–Salemal (*n* = 16) |
|---|---|---|---|
| Cl | 1740 ± 510 | 1660 ± 430 | 7240 ± 530 |
| SO$_4$ | 7070 ± 1340 | 5106 ± 888 | 8480 ± 392 |
| DOC | 4600 ± 930 | 7200 ± 2100 | 11,500 ± 400 |
| UV$_{245}$ * | 0.13 ± 0.45 | 0.28 ± 0.12 | 0.49 ± 0.02 |
| DIC | 18,500 ± 5300 | 16,690 ± 2180 | 15,100 ± 766 |
| Li | 2.63 ± 0.92 | 1.91 ± 0.17 | 2.74 ± 0.19 |
| B | 13.9 ± 2.5 | 10.0 ± 0.63 | 17.7 ± 1.64 |
| Na | 5866 ± 1471 | 4533 ± 296 | 8635 ± 566 |
| Mg | 5086 ± 803 | 4092 ± 308 | 4788 ± 236 |
| Al | 6.84 ± 1.96 | 11.56 ± 3.02 | 18.67 ± 2.28 |
| Si | 2987 ± 934 | 2905 ± 797 | 2440 ± 75 |
| P | 30.7 ± 14.7 | 49.030 ± 19.5 | 75.4 ± 5.7 |
| K | 1115 ± 183 | 972 ± 137 | 1571 ± 98 |
| Ca | 29,664 ± 3822 | 22,677 ± 2683 | 19,978 ± 914 |
| Ti | 3.27 ± 1.20 | 5.46 ± 1.84 | 8.80 ± 1.04 |
| V | 1.67 ± 0.3 | 1.37 ± 0.08 | 1.52 ± 0.03 |
| Cr | 0.14 ± 0.11 | 0.27 ± 0.12 | 0.30 ± 0.12 |
| Mn | 2.0 ± 1.48 | 18.0 ± 49.0 | 4.5 ± 0.5 |
| Fe | 59 ± 62 | 440 ± 475 | 664 ± 58 |
| Co | 0.04 ± 0.02 | 0.07 ± 0.1 | 0.07 ± 0.01 |
| Ni | 0.6 ± 0.2 | 0.9 ± 0.5 | 1.9 ± 0.1 |
| Cu | 1.7 ± 0.5 | 1.7 ± 0.2 | 2.3 ± 0.2 |
| Zn | 4.8 ± 3.2 | 3.0 ± 1.5 | 5.5 ± 3.2 |
| Ga | 0.01 ± 0.004 | 0.01 ± 0.002 | 0.01 ± 0.002 |
| Ge | 0.01 ± 0.003 | 0.007 ± 0.002 | 0.009 ± 0.001 |
| As | 1.4 ± 0.2 | 1.3 ± 0.5 | 1.5 ± 0.07 |

<div align="center">**Table 1.** *Cont.*</div>

| Descriptions | Tom–Vasyugan (*n* = 4) | Vasyugan–Irtysh (*n* = 11) | Irtysh–Salemal (*n* = 16) |
|:---:|:---:|:---:|:---:|
| Rb | 0.6 ± 0.09 | 0.8 ± 0.5 | 1.2 ± 0.09 |
| Sr | 165 ± 33 | 126 ± 13 | 123 ± 6 |
| Y | 0.04 ± 0.03 | 0.1 ± 0.05 | 0.3 ± 0.01 |
| Zr | 0.02 ± 0.02 | 0.06 ± 0.03 | 0.1 ± 0.01 |
| Nb | 0.002 ± 0.002 | 0.007 ± 0.005 | 0.02 ± 0.002 |
| Mo | 0.7 ± 0.08 | 0.4 ± 0.1 | 0.4 ± 0.03 |
| Cd | 0.01 ± 0.01 | 0.009 ± 0.009 | 0.01 ± 0.006 |
| Sb | 0.1 ± 0.02 | 0.1 ± 0.02 | 0.09 ± 0.004 |
| Cs | 0.001 ± 0.00 | 0.001 ± 0.000 | 0.002 ± 0.000 |
| Ba | 24 ± 3 | 22 ± 0.9 | 24 ± 0.7 |
| La | 0.03 ± 0.02 | 0.1 ± 0.04 | 0.2 ± 0.01 |
| Ce | 0.04 ± 0.03 | 0.2 ± 0.08 | 0.4 ± 0.02 |
| Pr | 0.01 ± 0.01 | 0.03 ± 0.01 | 0.06 ± 0.003 |
| Nd | 0.03 ± 0.03 | 0.1 ± 0.04 | 0.2 ± 0.02 |
| Sm | 0.01 ± 0.01 | 0.03 ± 0.01 | 0.06 ± 0.004 |
| Eu | 0.004 ± 0.002 | 0.008 ± 0.002 | 0.015 ± 0.001 |
| Gd | 0.01 ± 0.01 | 0.03 ± 0.01 | 0.06 ± 0.004 |
| Tb | 0.001 ± 0.001 | 0.004 ± 0.001 | 0.008 ± 0.000 |
| Dy | 0.006 ± 0.004 | 0.02 ± 0.008 | 0.05 ± 0.003 |
| Ho | 0.001 ± 0.001 | 0.004 ± 0.002 | 0.009 ± 0.001 |
| Er | 0.004 ± 0.003 | 0.01 ± 0.004 | 0.03 ± 0.002 |
| Tm | 0.000 ± 0.000 | 0.002 ± 0.001 | 0.004 ± 0.000 |
| Yb | 0.003 ± 0.002 | 0.01 ± 0.004 | 0.02 ± 0.002 |
| Lu | 0.000 ± 0.000 | 0.002 ± 0.001 | 0.003 ± 0.000 |
| Hf | 0.002 ± 0.001 | 0.008 ± 0.004 | 0.02 ± 0.002 |
| W | 0.03 ± 0.01 | 0.02 ± 0.007 | 0.01 ± 0.002 |
| Tl | 0.002 ± 0.001 | 0.001 ± 0.000 | 0.002 ± 0.000 |
| Pb | 0.07 ± 0.1 | 0.2 ± 0.1 | 0.2 ± 0.02 |
| Th | 0.001 ± 0.001 | 0.007 ± 0.004 | 0.02 ± 0.002 |
| U | 0.4 ± 0.06 | 0.3 ± 0.1 | 0.2 ± 0.03 |

<div align="center">* units are $cm^{-1}$.</div>

The distinction of the riverine solutes into three major groups depending on their latitudinal pattern was also confirmed by Spearman correlation coefficients between the element concentration and the latitude (Table S3 of the Supplementary Material). The elements of the first group exhibited quite strong ($\leq -0.90$, $p < 0.005$) negative correlations with latitude, whereas the elements of second group had $R_{Spearman} > 0.80$. The elements with a weak or non-systematic pattern showed statistically significant correlations ($p < 0.05$), but the $|R_{Spearman}|$ was typically below 0.8.

### 3.2. Major and Trace Elements in the Tributaries

The 11 tributaries sampled in this study exhibited highly contrasting behavior in both major and trace element concentrations. Similar to the main stem, this allowed the revealing of a distinct group of elements according to their latitudinal pattern. The southern

tributaries (upstream of the Irtysh) were enriched in DIC, major anions, alkali and alkaline-earth metals and U, whereas the northern tributaries exhibited much higher concentrations of Mn, Fe and Co (>5 times) and were sizably enriched in DOC, Cr, P, Zn, Cd, Nb and trivalent and tetravalent low soluble trace elements (Figure S3).

The Irtysh River played a governing role in the concentration pattern of the main stem, as it presented a sizable addition of many elements, mostly labile anions and oxyanions (DIC, Cl, $SO_4$, B, Mo, Sb and W), alkalis and alkaline-earth metals (Li, Na, K, Rb, Mg, Ca, Sr and Ba), some divalent transition metals (Mn, Co and Ni), U and DOC to the main stem, given its high discharge (49% of the Ob River at the confluence in 2016) and, most importantly, elevated concentration of elements (Figure 4). Some elements, however, exhibited a higher concentration in the northern part of the Ob River compared to the Irtysh River: Al, Fe, Cr, Zn, Ga, Nb, Ti, Zr, Hf and REEs (Figure 4). Presumably, these elements were additionally delivered to the Ob River main stem via multiple small-size tributaries, which were not sampled in the present work.

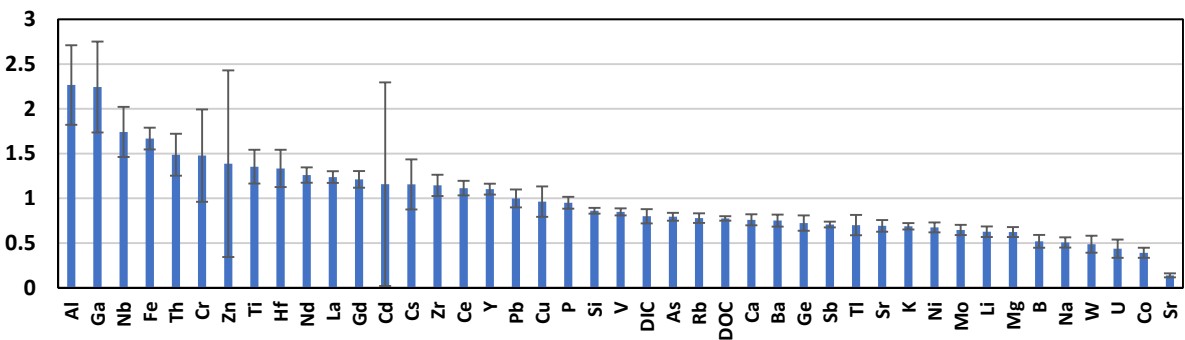

**Figure 4.** The ratio of the average (±s.d.) element concentrations of elements in the Ob River main stem downstream of the Irtysh (till the Ob mouth) to the concentration in the Irtysh River. Values above 1 indicate a sizable enrichment of the main stem in a given element by small tributaries in the north.

*3.3. Colloidal Status of Major and Trace Elements in the Ob River and Tributaries*

The dialysis procedure allowed a first-order assessment of the colloidal fraction (defined as the % of the element in the 1 kDa—0.45 μm fraction, divided by its total dissolved concentration in the <0.45 μm fraction). The percentage of colloidal fraction ranged from 0–10% (Cl, $SO_4$, DIC, Na, K, Li, B, Si, Mg, Ca and Mo) to 80–90% (trivalent and tetravalent hydrolysates), as listed in Table S4. In the northern part of the basin (downstream of the Irtysh), the distribution of the colloidal forms of elements among the tributaries and the main stem was quite homogeneous. This allowed identification of three main groups of solutes with respect to their colloidal status (Figure 5): (1) alkalis and alkaline-earth metals, major anions and Si and trace oxyanions (Mo, Sb and Ge) presenting between 0 and 20% of colloidal forms; (2) DOC, divalent transition metals (Ni, Cu and Cd), V, Cr, Cs, Tl, W and U, which were sizably impacted by colloids (20–70%) and (3) P, Fe, Mn, Co and all trivalent and tetravalent hydrolysates (Al, Ga, Y, REEs, Ti, Zr, Hf and Th), which were present essentially (>70–80%) in the colloidal form.

The DOC exhibited a remarkably homogeneous proportion of colloidal forms (42 ± 3%) in the main stem and most tributaries, including the Irtysh. However, the two most southern tributaries (Ket' and Tom') exhibited a much lower proportion of colloidal DOC, as well as of Fe, Co and Ni. Other colloid-affected elements (Al, Ti, P, V, Cr, As, Ga, Y and Zr) also demonstrated the lowest proportion of colloids in the southern tributaries (Irtysh, Ket and Tom). Uranium exhibited quite a particular colloidal pattern, with a gradual decreasing of the colloidal fraction from the north (70–90%) to the middle course, including the Irtysh (20 to 50%), and further decreasing in the two most southern tributaries, Ket' and Tom' (3%).

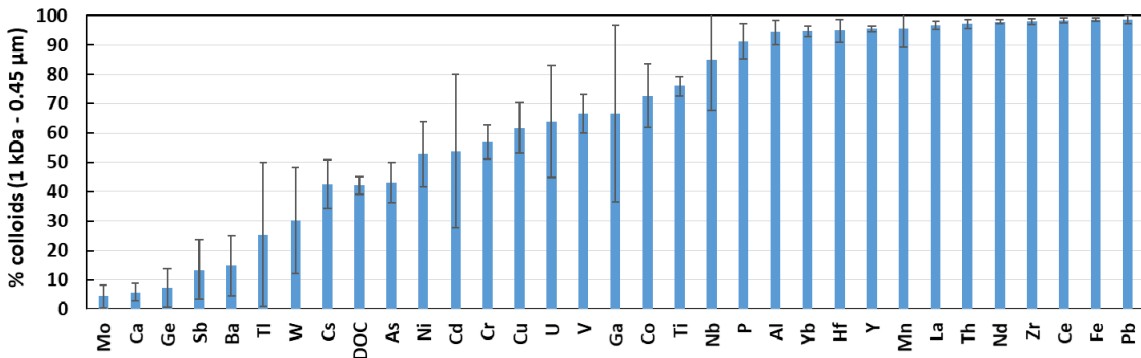

**Figure 5.** Proportion of the colloidal fraction of elements in the Ob River downstream of the Irtysh (4 points) and 3 tributaries (Pasaydeyakha, Poluy and Pitljar). Alkalis and alkaline-earth metals and oxyanions exhibited <10% of colloids and are not shown here.

It is noteworthy that the role of the largest tributary (Irtysh) in the colloidal pattern of the Ob River was not that significant: for most elements, their colloidal fraction in the Irtysh was not statistically different from that of the main stem and the downstream (northern) tributaries (Figure S4).

### 3.4. Testing the Landscape, Soil and Quaternary Deposits Control on Element Concentration

The main stem of the Ob River and its tributaries sampled in this study exhibited strong variability in the main landscape parameters, such as watershed size; mean annual air temperature (MAAT) and permafrost, forest, lake, bog and floodplain coverage, as well as the type of soil and Quaternary deposits. The main environmental parameters of the Ob River basin progressively evolved from the south to the north, which allowed testing of the impact of the climate and landscape on the element concentration in several selected points of the main stem. The latitude of the sampling point; the permafrost coverage and the proportion of bogs, lakes, floodplain, lacustrine and fluvio-glacial Quaternary deposits of the watershed strongly correlated ($p < 0.05$) with DOC, Fe, P, divalent metals (Mn, Co, Ni, Cu and Pb), Rb and low mobile lithogenic trace elements (Al, Ti, Cr, Y, Zr, Nb, REEs, Hf and Th) concentrations (Table S3A,B, Figures 6 and S5). Note that the impact of permafrost was especially strong in the northern part of the river basin. The pH and concentrations of soluble, highly mobile elements (DIC, $SO_4$, Ca, Sr, Ba, Mo, Sb, W and U) positively correlated with the proportion of forest, loess and fertile soils and eluvial, eolian, and fluvial Quaternary deposits on the Ob River watershed (Figures 6 and S5, where a number of the most important parameters, such as forest coverage (pH, Ca, Mo and U), floodplain area (DOC, P, Fe, Th) and watershed coverage by fluvio-glacial deposits (DOC, Al, Fe and La) are illustrated). Other landscape factors were of secondary importance for element control, yet they exhibited sizable correlations with particular elements. Examples are podzol soil coverage of the watershed that was positively linked to the concentration of Al, Cr, Ga, Th and saline soils that positively impacted the concentration of Cl, $SO_4$, Li, B, Na, K, As and Rb in the river water.

Pairwise correlations of the riverine element concentration with the landscape parameters of the watershed were further developed via a multi-parametric statistical approach (Principal Component Analysis). Considering both the main stem of the Ob River and its tributaries, two main factors were revealed, accounting for 41% and 13% of total variability, respectively (Figure 7). The F1 was presumably controlled by a northward increase in permafrost, floodplain, bogs, lakes and lacustrine deposits on the watersheds and acted on the DOC, Al, P, Ti, Fe, divalent metals, Rb, Cs, Nb and trivalent and tetravalent trace elements concentrations. The second factor (F2) included the Specific Conductivity, $SO_4^{2-}$, Li, B, Na, labile alkaline-earth metals (Mg, Sr and Ba), oxyanions (Mo, Sb and W) and oxyanions (As, Sb), which were impacted by southward-dominating forest coverage, loesses and eluvial and fertile soils. This factorial structure was found to be highly stable and preserved in the

general features for both the Ob River main stem and its tributaries, if treated separately (Figure S6).

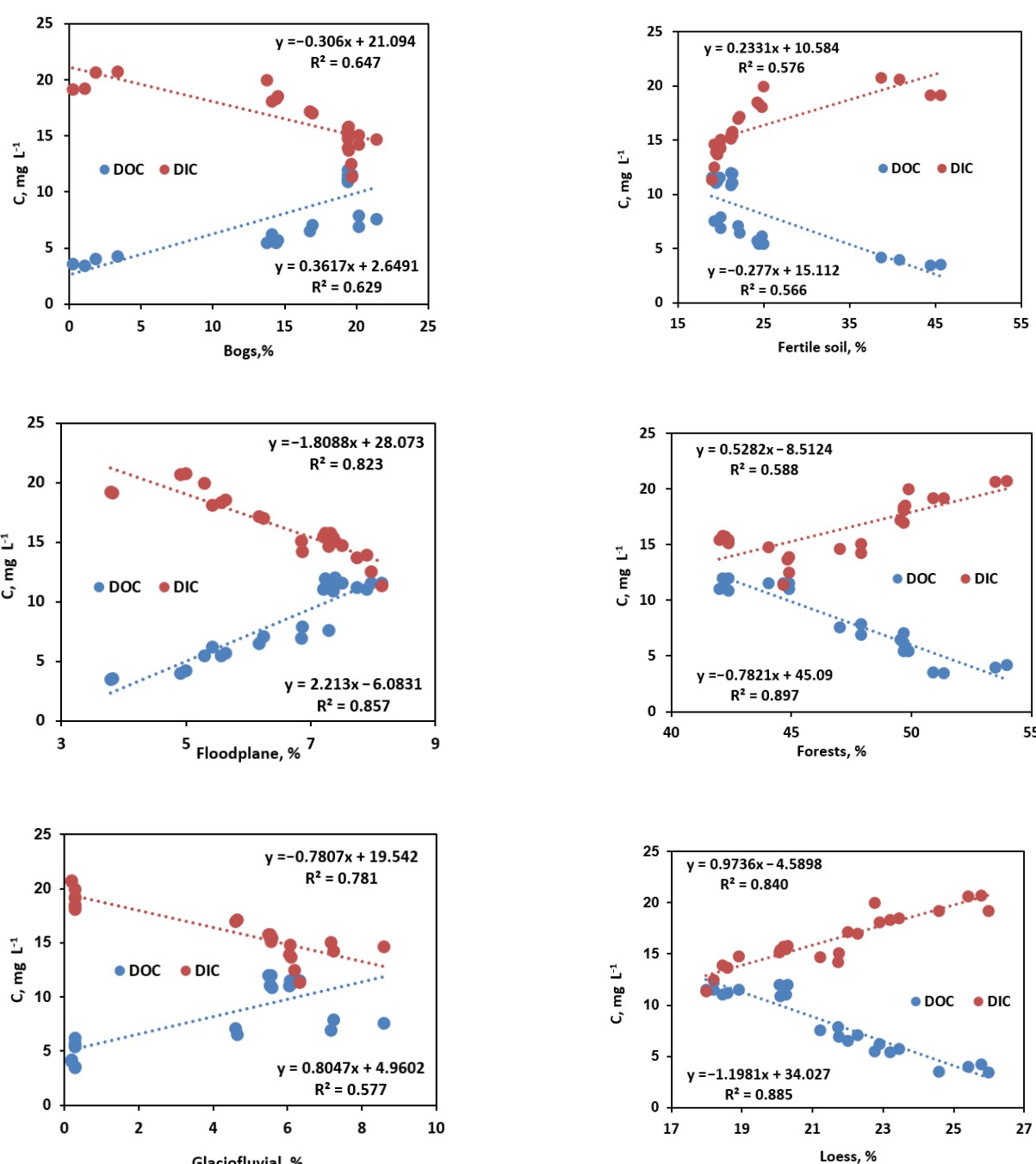

**Figure 6.** Examples of landscape factors, soil and Quaternary deposit control on the DOC and DIC concentrations in the Ob River. See examples of other factors and elements in Figure S5. A correlation matrix of environmental parameters of the watershed and riverine solutes is provided in Table S3.

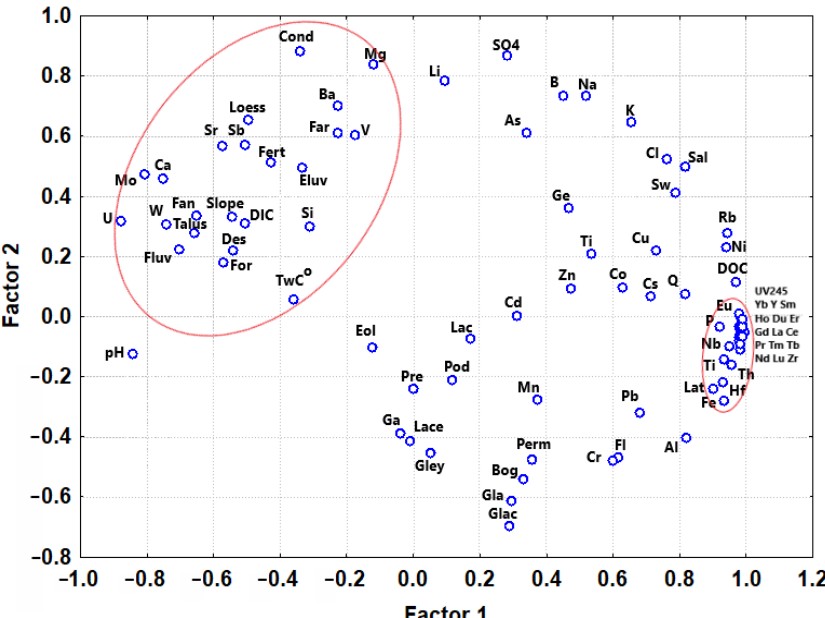

**Figure 7.** Results of the PCA treatment of all of the dataset (the Ob River main stem and tributaries), including the elementary composition of the river water and landscape parameters of the watersheds. The landscape parameters (% of watershed coverage) are abbreviated as follows: For, Forest; Fl, floodplain; Perm, permafrost; Far, farmland; Pre, pre-Quaternary deposits; Glac, glacial deposits; Slope, slopewash deposits, Des, deserptium (rock stream deposits); Lac, lacustrine and glacio-lacustrine deposits; Fan, fan-alluvial deposits; Gla, glacial deposits; Eol, eolian deposits; Pod, podsols; Sal, saline soils; Fert, fertile soils; TwC°, Water temperature; Cond, Specific Conductivity.

## 4. Discussion

*4.1. The Contrasting Spatial Distribution of DOC and Major and Trace Elements between Northern and Southern River Segment Is Due to the Latitudinal Pattern of Landscape, Soil and Quaternary Deposits*

The hydrochemical composition of the Ob River basin demonstrated several distinct groups of elements, defined according to the latitudinal patterns of their concentrations, similar to what was reported for small rivers of the WSL [31,36]. The DOC, organically bound metals (V, Cr, Mn, Fe, Co, Ni, Cu, Zn, Cd and Pb) and the majority of low soluble trace elements (Al, REEs, Nb, Ti, Zr, Hf and Th) exhibited a minimal concentration in the upper reaches of Ob and a maximal concentration in the northern, permafrost-affected zone. The northward increase in the concentration of these elements in both the main stem and tributaries may have originated from multiple factors. First, enhanced input of lithogenic low mobility elements in the northern part of the Ob River (permafrost zone) may have occurred due to a decrease of river connectivity, with deep and shallow groundwater and an increase in the surface flow. The surface drainage through forest litter together with surface runoff from surrounding bogs have led to river water enrichment in organic and organo-ferric and organo-aluminum (DOC, Fe, Al) colloids [30,34]. These colloids act as main carriers of divalent metals, insoluble trivalent and tetravalent trace elements, Pb, Cr and V in western Siberian rivers [27,28].

Another possible explanation for a northward increase in the concentrations of lithogenic and low mobile elements and a decrease in the concentrations of soluble labile elements lies in the dynamics of peat formation/decay across the territory, as recently suggested for small rivers of the WSL [36]. Because of ongoing recovery from the last glaciation, the WSL terrestrial ecosystems are not at the stationary stage: the growth of mires in the south leads to active accumulation of peat [40,41], whereas the permafrost peatlands in the north thaw and degrade [42–44]. Given that the WSL peat preferentially accumulates heavy metals (V, Cr, Zn and Pb) and trivalent ($TE^{3+}$) and tetravalent ($TE^{4+}$)

trace elements (Al, Y, REEs, Ti, Zr, Hf and Th) and is depleted in alkalis and alkaline-earths metals, As and Mo [45], it is possible that the riverine concentrations of trace elements inherit the element cycling between the growing/decaying peat and surface hydrological network.

At the same time, the obtained results did not completely confirm the initial hypothesis. We expected a northward decrease of the concentration of ground-water originated soluble, highly mobile anions, alkalis and alkaline earth metals. However, this was not observed for Cl, $SO_4$, Na and K. "Presumably, these elements bear the influence of salt soils of the Irtysh River, which flows through partially salty steppe and forest-steppe landscapes. Further to the north, the Ob River drains through paleo-marine deposits containing salt minerals (based on sedimentary cores available from extensive drilling of the territory [46]). This may enrich the main stem in highly mobile Na, Cl and $SO_4$.

It is important to note that unlike it was reported for water $pCO_2$ and molecular composition of riverine DOM (i.e., [17–19], the Irtysh River does not play a regulatory role of the lower Ob River water chemistry in terms of the concentrations of other major and trace elements, including organo-mineral colloids. Instead, the hydrochemical conditions of the Ob River are shaped by the integration of multiple small tributaries, the chemical composition of which gradually changes along the permafrost and landscape gradient, as is known for other river basins of western Siberia, such as Taz and Pur [36].

The concentration of riverine solutes was found to be strongly controlled by the abundance of bogs and lakes on the watershed and its floodplain coverage, for both the main stem and tributaries. The elements which positively correlated with wetlands reflect a leading role of both organic-rich soils and sediments of wetlands (unified here as bogs, lakes and floodplains) in DOC, P, K, divalent metals and insoluble elements mobilization (notably trivalent and tetravalent hydrolysates) from the watershed to the river. This presents a prominent contrast to the very small boreal catchments described in Northern Sweden, where the presence of wetlands at the watershed decreased the fluxes of dissolved metals from boreal forests to downstream [47–49]. According to these authors, such a decrease occurred due to a combination of low weathering in peat soils and the accumulation of organophilic metals in peat. The contrast between the northern Scandinavian and western Siberian settings is consistent with a possibility of peat degradation, rather than accumulation in northern part of the Ob River basin (downstream of the confluence with the Irtysh). This would lead to the enhanced release of trace metals to the streams and rivers. It is also possible that the presence of permafrost in the northern part of the WSL greatly shortens the water and element pathways between wetlands and rivers, compared to that in the non-permafrost regions of northern Sweden. The transport of trace metals released from mineral soils under the forest towards the river is therefore strongly enhanced by allochthonous organic matter originating from peat bogs and wetlands of the region.

Another landscape parameter is the presence of podzol soils at the watershed. Similar to glacial quaternary deposits, these soils lead to enrichment of the river water with low mobile, lithogenic elements, such as Al, Cr, Ga, REE, Hf and Th (see Table S3). These elements mark the presence of primary silicate minerals, often developed on felsic moraine, as has been shown in European boreal regions (i.e., ref. [50]).

In contrast to metals and low mobile trace elements, the concentrations of the DIC, alkaline-earth metals, oxyanions (Mo, As and W) and U decreased northward and were positively affected by the presence of forest, loess, eluvial, fluvial and fertile soils. On the one hand, this could reflect a northward decrease of the connectivity between groundwaters (enriched in these elements; see ref. [51]) and surface waters due to an increase in the permafrost coverage (i.e., [33,34]). On the other hand, a southward increase in watershed coverage by soils which are rich in these labile elements (loess and eluvial and eolian soils, containing carbonate concretions and fauna) may explain the preferential enrichment in these elements in the southern part of the Ob River watershed. Note that the eolian solid aerosol deposits in western Siberia are also enriched in these soluble elements [52]. It has

not been excluded that both factors (soils and groundwaters) were pronounced at the scale of such a large watershed.

Finally, the presence of saline soils in the southern part of the Ob basin produced a sizable enrichment of the river water in anions (Cl and $SO_4$), B, alkalis (Li, Na and K), which presumably originated from local surface salt deposits and evaporates.

### 4.2. Riverine Transport of the Trace Element Occurs in the Form of Organo-Ferric Colloids

All major and trace elements in the Ob River and its tributaries presented a colloidal pattern, which was similar to the one established for boreal and permafrost-affected zones of high latitudes [28,37,53–57]. Soluble, highly labile alkalis, anions and oxyanions were negligibly affected by colloids as they do not interact with organic matter or colloidal Fe/Al hydroxides and thus present in the form of simple ions or neutral molecules. The second group of colloidal-bound trace elements reflected their capacity to either complex with colloidal organic matter (divalent transition metals) or co-precipitate with organo-ferric colloids (trivalent and tetravalent hydrolysates).

A prominent feature of the major part of the Ob Basin is that the spatial variability of the colloid distribution among the tributaries and the main stem was rather low, and only two southern tributaries (Ket and Tym) presented a remarkable contrast to the rest of the Ob River basin. This presumably reflects the homogeneous physico-geographical and landscape parameters of the majority of the Ob River and its tributaries north of the Irtysh. The two southern tributaries exhibit less bogs and lakes in their watersheds and contain carbonate minerals in their base rocks (clays and silts with carbonate concretions, [30]). In accordance with previous studies of small rivers of the WSL [58], we hypothesize that the presence of carbonate rocks and decreasing the proportion of wetlands in the southern part of the Ob basin leads to (1) a decrease in the concentration of DOC and Fe, which can serve as main carriers of trace metals and (2) an increase in the DIC concentration and enhanced delivery of the ionic form of elements from the groundwater. The latter is facilitated by a much stronger connectivity between DOC-poor and DIC-rich groundwater and surface waters in the permafrost-free part of the Ob Basin [34].

An interesting and unexpected result was the extremely high proportion of colloidal Mn (83 to 100% in all samples except Ket (23%)). The typical colloidal fraction of Mn in the boreal and permafrost-affected surface waters (rivers, lakes and bog waters), including peat porewater, is between 80 and 20% [28,29], and the summer period usually accounts for the lowest proportion of colloidal Mn, due to the dominance of low molecular weight (<1 kDa) exometabolites that are capable of complexing divalent metals [27]. However, the lowest proportion of Mn colloids was observed only in the most southern tributary, which also exhibits a low proportion of colloidal OC. In these rivers, Mn is likely to be present as a divalent cation, originated from the groundwater, or complexed with autochthonous low molecular weight DOM. In contrast, the majority of Mn in all of the other tributaries and the main stem was tightly bound to colloids. We therefore hypothesize that the transport of Mn occurs in the form of high molecular weight, Fe-rich organic colloids in the main part of the Ob Basin. It is possible that a high proportion of bogs in the watersheds of all rivers north of the Vasyugan River provides such an enhanced colloidal transport. These Mn-rich colloids are likely to be generated at the interface of anoxic and oxic surface waters of the bogs and then delivered to the river via surface flow. Note that the enhanced riverine Mn transport in mire-affected regions has been reported in Northern Europe [55,59].

Another striking example of speciation control on element migration in the river water is that of the uranyl ($UO_2^{2+}$) ion. Uranium (VI) speciation in surface waters rich in DOM and Fe is largely controlled by high molecular weight organo-ferric colloids [28,60]. In the non-permafrost zone, these colloids are formed at the riparian/hyporheic zone of the river where the Fe(II) and U(VI)-rich groundwaters mix with oxygenated, organic-rich surface waters [37]. In the permafrost zone, these colloids originate from the surface flow over the forest floor and mire waters. Thus, in the northern part of the Ob River basin, elevated concentrations of both Fe and DOC provide suitable conditions for U transport

in the form of colloids (70–90% in the 1 kDa—0.45 μm form). Mires, which are highly abundant in the Ob basin north of the Ket tributary, are also capable of mobilizing uranium to the river water, as is known from other boreal settings [61]. In contrast, in the southern part of the basin, the groundwater contacting with the carbonate rocks of the basement or the loess soils are enriched in $HCO_3^-$. Such a chemical environment renders uranyl into soluble, non-colloidal carbonate complexes, as is known for other boreal regions which are impacted by carbonate rocks [62]. As a result, the proportion of the colloidal form of $UO_2(VI)$ progressively decreases southward and becomes as low as 3% in the two most southern tributaries (Ket and Tom), which exhibited the highest DIC and Ca and the lowest DOC and Fe concentration. Overall, the progressive southward decrease of the colloidal status of U reflects an increase in the connectivity between the surface and groundwaters of the WSL (i.e., [34,63]) and a decrease in the bog coverage of the river watershed in the same direction.

*4.3. Possible Impact of Landscape Changes on Element Concentration in the Ob River and Its Tributaries*

The unique geographical position of the Ob River, which traverses, from the south to the north, several distinct landscape zones and encompasses a large permafrost, MAAT and vegetation gradient, allows one to use a substituting space for time approach (i.e., ref. [64]) for foreseeing possible future changes in the river water hydrochemistry based on the contemporary pattern of riverine solutes. This approach has been efficiently used for small rivers of the WSL [30,31,33–35,58] but, to the best of our knowledge, never attempted for the Ob River main stem. The restrictions of this approach are the following: a lack of accounting for the time scale, necessary for the northern ecosystem to reach the new "more southern" state; ignoring the possible shift in the structure of the vegetation and soil microbial community, change in hydrologic seasons and the properties of ground-water [65]. Considering these restrictions, only a preliminary, first-order assessment of possible changes could be made.

Given the rather high similarity of riverine solutes in the Ob River and tributaries across different permafrost zones, north of the Irtysh (excluding the most southern part, south of Vasyugan), the shift in permafrost boundaries or the increase in the thickness of the active layer [66–69] are not expected to sizably impact the hydrochemical composition of rivers in the Ob basin, unless the connectivity between the deep underground and surface waters is modified (i.e., 10, 30). The latter may lead to the enhanced concentration of soluble highly mobile elements in the north, as has been demonstrated for the case of small WSL rivers of the Pur and Taz watersheds [27].

At the same time, the elements affected by the possible adsorption on the clay minerals underlying the peat deposits (first of all, DOC; see [70–74]) may modify their concentration in the northern part of the Ob basin. Globally, wetlands (bogs, lakes and floodplain zones) were the main controlling factor of the element concentration in the main stem and tributaries across the studied spatial gradient. Therefore, we believe that future changes in the WSL landscape, such as forestation of the northern part and a decrease in the proportion of bogs, will be all induced by changes in atmospheric precipitation, terrestrial productivity and duration of open-water seasons, i.e., [75]. However, quantification of the impact of these factors requires extensive ecosystem-level regional modeling [76], which goes beyond the scope of this study.

## 5. Conclusions

Based on the 3000 km north–south sampling transect of the Ob River main stem and its 11 tributaries, performed during the end of the spring flood—beginning of the summer baseflow period, this snapshot study of a large Arctic river dissolved (<0.45 μm) load revealed distinct physio-geographical control of the hydrochemistry of the river water. The land cover approach allowed testing of the control of the main physio-geographical parameters of the Ob River watersheds on element concentration along the main stem and among the tributaries. Bogs, floodplains, lake coverage and the permafrost pres-

ence on the watershed were found to be the main factors of the northward increasing in the concentration of DOC and low-soluble trace elements, which were present in the form of organic and organo-mineral colloids (Fe, Mn, Al, V, Cr, Co, Ni, Cu, Zn, Cd and Pb) and lithogenic trivalent and tetravalent TE (Al, Ti, Zr, Hf and Th) and REEs. These elements were also positively affected by the presence of glacial, lacustrine and fluvio-glacial Quaternary deposits. In contrast, soluble highly mobile alkalis, alkaline-earth metals, DIC, $SO_4$, B, Mo, Sb, As, W and U were positively affected by the presence of forest developed on loesses; fertile and saline soils and eluvial, fluvial and eolian Quaternary deposits, given that these substrates could contain soluble carbonate minerals and salt inclusions.

We revealed a high homogeneity in both the element concentration and colloidal status in the main stem of the Ob River and its tributaries located north of the confluence with the Irtysh. At the scale of the whole basin, the distribution of wetlands (bogs, floodplains, lakes and lacustrine deposits) was the dominant factor defining the elementary pattern in waters of the Ob River. In the northern part of the basin, the permafrost coverage exhibited sizable control on riverine solutes. As such, in case of drastic environmental changes in the WSL territory (permafrost boundary shift, active layer depth increase, vegetation coverage and precipitation regime), the changes in distribution of bogs, lakes and forest, might become the governing factors of modifications in the river hydrochemical regime.

**Supplementary Materials:** The following are available online at https://www.mdpi.com/article/10.3390/w13223189/s1, Figure S1: Latitudinal dependence of Cl, $SO_4$, Na, K and Pb concentration in the main stem of the Ob River; Figure S2: Main stem concentrations of elements which do not show any particular pattern with the latitude: Si, Mg, Cu, V, As and Sb; Figure S3: A histogram of the elemental ratio in 2 northern tributaries (downstream of the Irtysh) to 8 southern tributaries (upstream of the Irtysh) of the Ob River; Figure S4: Proportion of the colloidal (1 kDa—0.45 μm) fraction of DOC, Ca, Fe, Mn, Al, Cu, U, La and Th in the main stem (blue box plot column) and the tributaries of the southern (blue circles) and northern (red circles) part of the Ob Basin; Figure S5. Examples of major and trace element concentration with landscape parameters of the Ob River main stem and tributaries; Figure S6. Results of PCA treatment of the solute data and watershed characteristics (separately, Ob main stem and tributaries). Table S1: List of the sampled sites at the main stem of the Ob River and its tributaries. Table S2: Main landscape parameters and genetic type of the Quaternary deposits (% of the watershed coverage) of the tributaries and several key points at the Ob River main stem; Table S3: A correlation matrix of the element concentration in the Ob main stem tributaries and landscape coverage (%) of the watersheds; Table S4: Proportion of the colloidal fraction of elements in the Ob River downstream of the Irtysh (4 points) and 7 tributaries.

**Author Contributions:** I.K. and O.S.P. designed the study and wrote the paper; I.K., L.G.K. and S.N.V. performed sampling, analysis of major cations and their interpretation; I.P.S. and O.V.D. were the leaders of the ship expedition and provided data interpretation; L.S.S. was in charge of DOC, DIC and anion measurements and their interpretation; L.G.K., I.K., U.S. and R.S.V. provided GIS-based interpretation, mapping and identification of river watersheds. S.N.K. provided landscape characterization of river watersheds. All authors have read and agreed to the published version of the manuscript.

**Funding:** This research was funded by the Russian Scientific Foundation, RSF grant No 18-17-00237_P and grant No 21-77-30001 to IS, as well as by the Russian Foundation for Basic Research, RFBR grants No 19-55-15002, 20-05-00729_a, and RSF grant No 18-77-10045 for field work and a Belmont Forum VULCAR-FATE grant for some laboratory analyses. The APC was funded by O.S. Pokrovsky.

**Institutional Review Board Statement:** Not applicable.

**Informed Consent Statement:** Not applicable.

**Data Availability Statement:** All obtained data are published in ref. [39].

**Acknowledgments:** The study was partly carried out using the research equipment of the Unique Research Installation "System of experimental bases located along the latitudinal gradient" TSU, with financial support from the Ministry of Education and Science of Russia (RF—2296.61321X0043, agreement No. 075-15-2021-672). Furthermore, this study was supported by the Ministry of Education and Science of Russia (grant No 121-021-500057-4).

**Conflicts of Interest:** The authors declare no conflict of interest.

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
