# Peer review of "Landscape, Soil, Lithology, Climate and Permafrost Control on Dissolved Carbon, Major and Trace Elements in the Ob River, Western Siberia"

_water, doi:10.3390/w13223189_

Round 1

Reviewer 1 Report

In the reviewed manuscript, the authors investigate the influence of landcover, climate and permafrost on the water chemistry of the Ob river. The article in well written and present important information on the composition of a major Arctic river and identify the most important factors that influence the water composition, with special emphasis on DOC. I recommend the publication of the manuscript after minor revision.

Please find bellow some comments:

  1. Please verify figure 2 as each chart seems to be somehow truncated.
  2. L217-219 please verify the sentence.
  3. L228 please correct the typo: pattern
  4. L232 please define TE
  5. L321 please define MAAT

Author Response

In the reviewed manuscript, the authors investigate the influence of landcover, climate and permafrost on the water chemistry of the Ob river. The article in well written and present important information on the composition of a major Arctic river and identify the most important factors that influence the water composition, with special emphasis on DOC. I recommend the publication of the manuscript after minor revision.

Please find bellow some comments:

  1. Please verify figure 2 as each chart seems to be somehow truncated.

Response: There is certainly a problem of format transferring. This figure is just fine in our word file and the submitted pdf version, but the MDPI word file prepared based on our manuscript indeed produced some non-desired shifts in symbols and labelling of this figure. We made sure that the revised version contains the correct figure format and we inserted this figure in the text as a picture without changing its original format.

  1. L217-219 please verify the sentence.

Response: We thank the reviewer for pointing out some inconsistency in this sentence and we carefully revised it, via diving the main message into three independent sentences.

L228 please correct the typo: pattern - Fixed.

L232 please define TE. Trace elements, specified.

L321 please define MAAT. Mean Annual Air Temperature, explained.

Reviewer 2 Report

It is interesting and useful that the authors have investigated influence of Landscape, soil, lithology, climate and permafrost on dissolved carbon, major and trace elements in the Ob River, western Siberia. In total, the MS was written sound. Hence, it is recommended to be published after some revisions.

  1. “Abstract and Conclusion” is need to be shorten for;
  2. Figure 2 did not presented right, and cannot be seen;
  3. Delete one “in” in the caption of Table 1.

Author Response

It is interesting and useful that the authors have investigated influence of Landscape, soil, lithology, climate and permafrost on dissolved carbon, major and trace elements in the Ob River, western Siberia. In total, the MS was written sound. Hence, it is recommended to be published after some revisions.

  1. “Abstract and Conclusion” is need to be shorten for;

Response: We agree with this remark and removed 5 lines of text from the Abstract and 6 lines from the Conclusions.

2. Figure 2 did not presented right, and cannot be seen;

Response: There is certainly a problem of format transferring. This figure is just fine in our word file and the submitted pdf version, but the MDPI word file prepared based on our manuscript indeed produced some non-desired shifts in symbols and labelling of this figure. We made sure that the revised version contains the correct figure format and we inserted this figure in the text as a picture without changing its original format.

3. Delete one “in” in the caption of Table 1.

Response: Fixed; thanks for catching this!

Reviewer 3 Report

  1. The article is on a good topic, but it needs a small checking of its grammar and clarity.
  2. Figure 2 is not clear for reading.
  3. Please emphasize the representativeness of more than 20 sampling points for the 2,952 km river.

Author Response

It is interesting and useful that the authors have investigated influence of Landscape, soil, Comments and Suggestions for Authors

  1. The article is on a good topic, but it needs a small checking of its grammar and clarity.

Response: In the revised version, we thoroughly verified English grammar and spelling and we corrected a number of sentence for clarity. In many cases, we shortened the phrases to avoid presenting multiple ideas in a single sentence.

2. Figure 2 is not clear for reading.

Response: There is certainly a problem of format transferring. This figure is just fine in our word file and the submitted pdf version, but the MDPI word file prepared based on our manuscript indeed produced some non-desired shifts in symbols and labelling of this figure. We made sure that the revised version contains the correct figure format and we inserted this figure in the text as a picture without changing its original format.

3. Please emphasize the representativeness of more than 20 sampling points for the 2,952 km river.

Response: The reviewer made a good point here. Several lines of arguments allow us to argue that the sampling scheme is adequate for assessing the hydrochemical variations in the Ob River main channel. First, The 20 sampling points were semi-equally distributed over the main channel of the Ob River, making it one full sampling each 150 km. For such a large river system, a step of 150 km river length is a reasonable compromise for representability and feasibility of sampling. Second, unlike many other large rivers of the world which drain territories of variable lithology (Yenisey, Lena), diverse and contrasting relief, anthropogenic pressure and vegetation coverage (Great Asian rivers, Mississippi, European rivers), the Ob River drains highly homogeneous (in terms of lithology, landscape runoff and vegetation) Western Siberia Lowland, which is probably one of the most homogeneous (from the view point of physio-geographical parameters) territory of the world. And third, in our very recent study of CO2 concentration and emission from the main stem of the Ob River (Karlsson et al., 2021, Nature Comm.) we demonstrated highly homogeneous CO2 pattern, weakly linked to variable tributaries and exhibiting two main sectors of rather stable concentrations, south of the confluence with Irtysh, and the northern part downstream the confluence. The hydrochemical features of the Ob River assessed in the present study are basically consistent with this spatial pattern. Taken together, the 20 sampling of the main stem covering both northern and southern parts of the Ob River were representative for the studied river basin. We added an explicatory sentence to the revised version (section 2.1)